# Emerging Indications for Neoadjuvant Systemic Therapies in Cutaneous Malignancies

**DOI:** 10.3390/medsci12030035

**Published:** 2024-07-23

**Authors:** Domingos Sávio do Rego Lins Junior, Beatriz Mendes Awni Cidale, Ana Zelia Leal Pereira, Jacqueline Nunes de Menezes, Eduardo Bertolli, Francisco Aparecido Belfort, Rodrigo Ramella Munhoz

**Affiliations:** 1Oncology Center, Hospital Sírio-Libanês, São Paulo 01308-050, Brazil; domingos.lins@hsl.org.br (D.S.d.R.L.J.); ana.zpereira@hsl.org.br (A.Z.L.P.); rodrigo.rmunhoz@hsl.org.br (R.R.M.); 2Cutaneous Malignancies and Sarcoma Group, Hospital Sírio-Libanês, São Paulo 01308-050, Brazil; jacqueline.menezes@slserv.com.br (J.N.d.M.); bertollimed@gmail.com (E.B.); francisco.belfort@slserv.com.br (F.A.B.)

**Keywords:** immunotherapy, neoadjuvant therapy, melanoma, squamous cell carcinoma, basal cell carcinoma, Merkel cell carcinoma

## Abstract

Patients with cutaneous malignancies and locoregional involvement represent a high-risk population for disease recurrence, even if they receive optimal surgery and adjuvant treatment. Here, we discuss how neoadjuvant therapy has the potential to offer significant advantages over adjuvant treatment, further improving outcomes in some patients with skin cancers, including melanoma, Merkel cell carcinoma, and cutaneous squamous-cell carcinoma. Both preclinical studies and in vivo trials have demonstrated that exposure to immunotherapy prior to surgical resection can trigger a broader and more robust immune response, resulting in increased tumor cell antigen presentation and improved targeting by immune cells, potentially resulting in superior outcomes. In addition, neoadjuvant approaches hold the possibility of providing a platform for evaluating pathological responses in the resected lesion, optimizing the prognosis and enabling personalized adaptive management, in addition to expedited drug development. However, more data are still needed to determine the ideal patient selection and the best treatment framework and to identify reliable biomarkers of treatment responses. Although there are ongoing questions regarding neoadjuvant treatment, current data support a paradigm shift toward considering neoadjuvant therapy as the standard approach for selecting patients with high-risk skin tumors.

## 1. Introduction

The term skin cancer encompasses different entities that arise from healthy skin tissues and can be classified according to their histology and molecular aspects. The most common subtypes are cutaneous squamous cell carcinoma (cSCC) and basal cell carcinoma (BCC), as well as less common, yet challenging, subtypes, including melanoma and Merkel cell carcinoma (MCC) [1]. These cancers are responsible for more than 1.5 million new cases globally every year, resulting in a prevalence of over 7 million cases worldwide [2].

Although the mortality rates associated with skin tumors are not as high as those of some other types of cancer, this is a highly heterogeneous group in terms of tumor biology and long-term outcomes, and while the mortality rate for BCC remains low, more aggressive subtypes of cutaneous malignancies, including melanoma and MCC, frequently present with locoregional or distant metastases, resulting in an ominous prognosis [3,4,5].

The general treatment approach in the localized setting is based on the surgical removal of the primary lesion with an adequate margin and sentinel lymph node biopsy (SLNB), when indicated. Considering the recurrence rate after the surgical procedure, modalities of adjuvant therapies based on immune checkpoint inhibitors or targeted therapies can be considered in select scenarios [6,7,8,9]. The risk of recurrence varies according to tumor subtypes, being more common in high-risk stage II and stage III melanomas and in MCC with positive surgical margins or nodal involvement [5]. In patients with cSCC, the recurrence rate is higher in those who present high-risk tumor characteristics, such as the involvement of large or named nerves and having positive margins or extensive perineural involvement [3,8], and the use of adjuvant treatment is justified in minimizing the risk of recurrence and improving long-term outcomes.

In cutaneous melanoma, current guidelines still consider adjuvant treatment with immunotherapy as a standard of care. Despite the lack of uniform data confirming an improvement in overall survival (OS), adjuvant treatments, either in the form of target-therapies or immune-checkpoint blockers, resulted in gains in progression-free survival (PFS) and distant-metastasis-free survival (DMFS) in distinct pivotal trials [10,11,12,13,14,15]. This scenario will likely change in the near future, with the recently published data of the NADINA trial showing a significant improvement in event-free survival in patients who received the neoadjuvant treatment compared to those who received the adjuvant therapy [16]. In those with stage III disease, ipilimumab administered at a dose of 10 mg/kg every three weeks was the first monoclonal antibody targeting immune checkpoints tested in this setting; the randomized phase III EORTC 18071 trial resulted in significant gains in 7-year recurrence-free-survival (RFS) (39.2% vs. 30.9%; HR 0.75; *p*: 0.0004) and 7-year DMFS (44.5% vs. 36.9%; HR 0.76; *p*: 0.0018) versus the placebo; of note, this trial was the sole recent study to demonstrate a gain in 7-year OS (60.0% vs. 51.3%; HR 0.73; *p*: 0.0021) [10]. Subsequently, nivolumab and pembrolizumab were compared to active control arms or placebos across different randomized studies, also resulting in an increase in RFS and DMFS, with relative risk reductions of approximately 40–45% but without a significant difference in OS [11,12,13,14]. Similarly, BRAF/MEK inhibitors for those with stage III disease carrying an activating mutation in the BRAF V600 gene are currently approved for clinical use, based upon the results of the COMBI-AD study, in which the combination of dabrafenib and trametinib also led to improved outcomes compared to the placebo in patients undergoing adjuvant treatment after the surgical resection of stage IIIA to stage IIIC melanoma [15]. More recently, the indication for adjuvant treatment has been extended to those presenting with high-risk stage II disease based on the results of two randomized controlled trials that compared pembrolizumab or nivolumab to placebo in stage IIB and IIC melanoma patients, with relative risk reductions in RFS and DMFS similar to those observed in stage III disease [17].

While still considered investigational, the use of adjuvant approaches has also been addressed in distinct cutaneous malignancies, including MCC and cSCC [18,19]. In non-melanoma skin cancer (NMSC), although immunotherapy treatment strategies for metastatic disease have been successful, the role of adjuvant systemic therapy after complete resection, particularly in those with high-risk NMSC, is unclear, with distinct phase III studies underway [20,21,22,23].

Although encouraging, these data present a reality in which close to half of patients still experience recurrence during the long-term follow-up, with a considerable amount of distant metastasis, raising questions about other strategies that could potentially reduce these numbers; it is also important to note that a significant proportion of patients develop disease recurrence early after surgery, before the commencement of any adjuvant treatment, contributing to a high proportion of patients who require salvage therapies [11,13,24].

More recently, neoadjuvant therapy has been gaining ground in resectable high-risk cutaneous malignancies because of its potential benefits, which include the following: tumor reduction, with subsequent less morbid surgeries with better outcomes; testing in vivo responses to systemic therapy; providing prognostic data according to pathological responses; the customization of adjuvant treatment based on the response to neoadjuvant therapy; the elimination of micrometastases, allowing for the study of the tumor microenvironment; and the early determination of efficacy signals, potentially enabling a more rapid platform for drug development [25]. Recently, both the complete pathological response (pCR, defined as no residual viable tumor) and major pathological response (MPR, defined as ≤10% residual viable tumor assessed on biopsy) rates have been suggested as strong surrogate markers for both RFS and OS in patients with high-risk melanoma, further affirming the benefit of using neoadjuvant treatment in selected patients [25].

Herein, we report on currently approved neoadjuvant therapy strategies for advanced skin cancers, the available evidence in respect of their application, future perspectives, and the challenges associated with their use.

## 2. Methods

We searched for current evidence on neoadjuvant therapy in international medical databases, including Pubmed, ClinicalTrial.gov, and Scielo. Studies involving adult subjects were included. There was no limitation regarding the publication year of the studies. A review of the literature was carried out from 1 January 2010 to 15 January 2024. The descriptors (‘Melanoma’ OR ‘squamous cell carcinoma’ OR ‘basal cell carcinoma’ OR ‘Merkel cell carcinoma’) AND neoadjuvant therapy OR Immunotherapy OR target therapy were used. Relevant articles on this topic were selected. References of the main articles and abstracts of the leading congresses in the area were also included to ensure all the available data were utilized to conduct a literature review that thoroughly examined the current scenario of perioperative treatment in skin tumors, elucidating the biological rationale for neoadjuvant treatment and its potential benefits.

## 3. Rationale for Neoadjuvant Therapy

The use of neoadjuvant therapy has several potential advantages over adjuvant approaches, including the opportunity to assess the pathological response in the dissected lesion, optimized prognostication, and the chance to adopt personalized adaptive management, which may further improve the outcomes of patients with high-risk resectable cutaneous malignancies [26].

Both preclinical and translational studies provide a strong rationale supporting these premises, particularly in respect of immunotherapy [26].

In animal models involving orthotopically implanted 4T1.2 mammary carcinoma cells in BALB/c mice, representing a human triple-negative breast cancer model, neoadjuvant treatment with anti-PD-1 combined with anti-CD137 exhibited statistically superior OS rates compared to those of the corresponding adjuvant treatment. Mechanistically, this observed benefit relied on INF-gamma production and the activation of CD8, CD4, and NK cells. Moreover, the authors demonstrated an increase in tumor-specific CD8+ T cells in the bloodstream four days after the neoadjuvant treatment, serving as a predictive marker for the efficacy of immunotherapy [26]. Similar results were also observed in an analogous model using mice implanted with B78 melanoma cells [27].

In this context, the benefit of neoadjuvant immunotherapy was contingent upon the activation of tumor-residing Batf3-dependent conventional type 1 dendritic cells. These dendritic cells, in turn, triggered the generation of tumor-specific T-cell responses, which positively impacted survival [28]. Indeed, it has been demonstrated that exposure to an immune checkpoint blockade before the surgical resection of melanoma, in the setting of a potentially immunogenic tumor burden yet within a less immunosuppressed microenvironment, may elicit a broader and stronger immunologic response [26,27,28,29].

The use of immunotherapy prior to surgical resection has been associated with a marked increase in intratumoral T cell expansion, as well as with a broader T cell receptor clonality. These factors, combined with the presence of tertiary lymphoid structures and a viable tumor assessment, are also important for prognosis [30,31,32]. The clinical response to immunotherapy and immune reinvigoration in the tumor can occur rapidly, with some cases of a complete or major pathological response being detected three weeks after a single dose of Anti-PD 1 [33].

In addition, the dynamic characterization of the tumor microenvironment during neoadjuvant treatment also offers the possibility of an optimized assessment of response predictors to the therapy, based upon the evaluation of biomarkers that include the mutational burden (TMB), PD-L1 expression, inflammatory genes expression profiles (GEPs), and circulating tumor DNA (ctDNA), which may prompt adaptive treatment strategies, as well as the possibility of exploring new potential biomarkers and targets that may contribute to the understanding of mechanisms of resistance [25].

## 4. Current Scenario

### 4.1. Melanoma

There have been significant advances in the melanoma treatment landscape since the introduction of ipilimumab in 2011, with several other drug approvals, in both metastatic and adjuvant settings, translating into substantial survival gains and dramatic improvements in patients’ long-term outcomes [34]. More recently, the neoadjuvant setting has emerged as a field of possible new gains [26,35,36,37].

Since 2016, preclinical evidence suggests that neoadjuvant therapy with immune checkpoint inhibitors may have an important role in the treatment of melanoma [26]. In an effort to demonstrate the clinical relevance of these data, Huang et al. tested the effect of a single dose of a neoadjuvant PD-1 blockade in patients diagnosed with stage III/IV resectable melanoma, with 8 out of 27 patients achieving a complete or major pathological response. Notably, individuals with a major pathological tumor response exhibited a 100% DFS rate at 24 months of the follow-up, while patients lacking robust pathological responses at surgery faced close to a 50% risk of recurrence, despite receiving adjuvant therapy. This underscores the potential of neoadjuvant therapy to improve clinical outcomes for resectable melanoma [33]. To test the feasibility of this approach, in 2018, phase 1b of the OpACIN study included patients with stage III melanoma, randomizing them into two arms, one with adjuvant therapy and the other with neoadjuvant therapy, both with ipilimumab in combination with nivolumab. This resulted in a pathological response in 78% (3 out of 9 being pCRs, 3 being near pCRs, and 1 being a partial pathological response [pPR]) of the patients who received neoadjuvant treatment, in addition to a greater quantity of T cell clones residing in the tumor [36]. In a distinct phase 2 study, Amaria et al. sought to scrutinize the optimal regimen in this context by comparing neoadjuvant treatment with nivolumab against the combined administration of ipilimumab with nivolumab in a cohort of 23 patients diagnosed with high-risk resectable melanoma. The combination of ipilimumab and nivolumab exhibited elevated response rates (overall response rates [ORR] 73%, pathological complete response [pCR] 45%), albeit accompanied by significant toxicity (73% incidence of grade 3 treatment-related adverse events [trAEs], the most common being transaminitis (27%), colitis (18%), hyperthyroidism (9%), pneumonia (18%), arthralgias (9%), myositis and/or myalgias (9%), and electrolyte abnormalities (hypokalemia 9%, hyponatremia 18%)). Conversely, the administration of nivolumab as a monotherapy resulted in modest responses (ORR 25%, pCR 25%) coupled with low toxicity (8% incidence of grade 3 trAEs, with the only grade 3 trAE for this group being tumor-related pain) [38].

Subsequently, the phase II OpACIN-neo study corroborated these findings, resulting in an objective radiologic response rate of 57% and a pathological response rate of 77% (including 57% with a pCR) in patients undergoing neoadjuvant treatment, with the optimal neoadjuvant arm comprising two cycles of ipilimumab 1 mg/kg plus nivolumab 3 mg/kg. This was associated with 20% of grade 3 and 4 toxicities, thus being better tolerated than the standard dosing regimen, maintaining a high response rate, and making this schedule suitable for broader clinical use [37].

Further analysis demonstrated that high levels of interferon (IFN-gamma) and tumor mutation burden (TMB) were directly related to a partial pathologic response rate and lower long-term recurrence [39]. Further compelling evidence supporting the use of neoadjuvant treatment was provided by the SWOG 1801 trial. This phase II study randomized 313 patients with stage III or IV resectable melanoma, who received three doses of neoadjuvant pembrolizumab, followed by surgery, and subsequently fifteen cycles of the standard adjuvant treatment, consisting of adjuvant pembrolizumab or eighteen doses of pembrolizumab. The 2-year event-free survival (EFS) rates were greater in the neoadjuvant arm than in the control arm (72% versus 49%, *p* = 0.004), along with excellent tolerability (the incidence of adverse events grade ≥ 3 was 12% versus 14%). Of the 105 patients analyzed by a central pathologic center, 53% achieved an MPR (38% with pCR) [40]. More recently, the results were updated, and those patients who achieved an MPR had an RFS of 88% after 24 months. For those with no MPR, the 24-month RFS was 80%, still an interesting number. Even in those who did not achieve an MPR, the high RFS reaffirms that the neoadjuvant strategy should be considered in all patients with an indication and that perhaps MPR is not such a reliable marker, requiring further refinement [40].

The phase II trial, NeoPeLe, evaluated the addition of lenvatinib to pembrolizumab in the neoadjuvant treatment of stage III nodal melanoma, followed by a lymph node dissection and, subsequently, forty-six applications of pembrolizumab. Of the 20 patients who were randomized, 40% achieved pCR, a higher rate than in the studies of anti-PD1 in monotherapy, with an overall response ratio of 35%. These data were encouraging, and the trial investigations are ongoing [41].

Another combination evaluated in this setting was the association of anti-PD-1 with anti-lag-3. Patients received two neoadjuvant cycles of nivolumab and relatlimab every four weeks, followed by surgical resection and ten adjuvant applications of the combination. Of the 30 patients enrolled, 57% achieved pCR, and the 1- and 2-year RFS survival rates were 100% and 92%, respectively, for patients with any pathological response [42]. Subsequent analyses revealed a significantly higher number of tumor-infiltrating CD8+ T cells, an increased expression of PD-L1 tumor cells, a higher rate of T cell clonality, and higher levels of lymphoid markers in responding patients compared to non-responders [43]. 

An additional combination explored involved the use of anti-PD-1 pembrolizumab in combination with anti-TIGIT vibostolimab. This approach was evaluated in comparison to two distinct cohorts of patients with stage III melanoma. One group received pembrolizumab combined with gebasaxturev (coxsackievirus A21), while the other underwent pembrolizumab monotherapy prior to resection, followed by adjuvant pembrolizumab for one year. The ORR was 50% in the Pembrolizumab + Vibostolimab group, 32% in the Pembrolizumab + Gebasaxturev group, and 27% in the monotherapy group. The pCR rates were 38%, 28%, and 40% respectively, while the near pCR rates were 12%, 12%, and 7%. RFS was not reached in either arm, and the EFS rates at 18 months were 81%, 61%, and 79%, respectively. Treatment-related grade 3/4 adverse events (AEs) occurred in 8%, 24%, and 7% of the groups, with no instances of treatment-related grade 5 AE [43].

There are also phase II studies using targeted therapy with neoadjuvant dabrafenib and trametinib for 8 weeks, followed by surgical resection and adjuvant treatment with the same combination, in patients with stage III melanoma or resectable oligometastases carrying mutations in the BRAF V600E or V600K gene. The results were similar to those of those using neoadjuvant immunotherapy, showing around 50% pCR; however, greater toxicities required pauses during treatment to manage them [44,45]. 

In order to better analyze the associations between the pathological response, recurrence-free survival (RFS), and overall survival (OS) with neoadjuvant therapy in melanoma, the International Neoadjuvant Melanoma Consortium (INMC) pooled data from six clinical trials of anti-PD-1-based immunotherapy or BRAF/MEK-targeted therapy. In this study, of the 192 patients included, 141 had received immunotherapy (104, a combination of ipilimumab and nivolumab; 37, anti-PD-1 monotherapy), and 51 had received targeted therapy. In patients with pCR, near pCR, or pPR with immunotherapy, very few relapses had been seen after two years (RFS 96%), with no melanoma-related deaths in this period, while in those treated with targeted therapy, even achieving pCR, the RFS of 2 years was only 79% and the OS was 91%. Such data point to better outcomes with the use of immunotherapy in the neoadjuvant setting [39].

Another phase II study explored the concomitant combination of dabrafenib plus trametinib and pembrolizumab in patients with stage III resectable melanoma with a BRAF V600E mutation. The NeoTrio results showed a high pCR rate (62.5%) in the combination group, but this was associated with high toxicity [46]. 

In conclusion, there are consistent data supporting the use of neoadjuvant immunotherapy, but more studies are required to define the ideal indication for this type of targeted therapy [46]. The current evidence for the use of neoadjuvant treatment in melanoma is summarized in Table 1.

Despite the results of SOWG S1801, phase 3 trials were still lacking in neoadjuvant settings to lead to formal approval in clinical practice. This all changed with the recent publication of the NADINA trial, a phase 3 study that assigned patients with stage III resectable melanoma and at least one positive lymph node to one of two arms up-front surgery followed by adjuvant therapy with 12 courses of nivolumab (480 mg) every four weeks against preoperative ipilimumab (80 mg) plus nivolumab (240 mg) every three weeks for two cycles followed by total lymph node dissection and adjuvant therapy in the experimental arm was guided by depth of response. Patients who achieved a major pathologic response received no additional treatment. In contrast, the others received 11 cycles of adjuvant nivolumab or, in case they had *BRAF*-mutated disease, adjuvant dabrafenib plus trametinib for 46 weeks.

After a median follow-up close to 10 months, the 12-month event-free survival rate was 83.7% in the neoadjuvant arm and 57.2% in the adjuvant arm (hazard ratio [HR] = 0.32; *p* < 0.0001). The 12-month event-free survival by pathological response subanalysis shows us that patients with complete and near-complete responses achieve event-free survival rates of 95.4% and 94.1%, respectively. While this rate was 76.1% for partial responders, and for nonresponders, only 57.0%. Regarding adverse events, The NADINA observed a 29.7% rate of nonsurgical-related grade ≥ 3 toxicities in the neoadjuvant arm against 14.7% in the control arm [16]. Long-term outcomes, particularly looking into OS, still need to be determined.

### 4.2. Cutaneous Squamous-Cell Carcinoma

Although most patients with cSCC present with early-stage disease and can be treated with surgery alone, a small portion of them present with locally advanced disease or with adverse histopathological characteristics, requiring the consideration of adjuvant radiotherapy and, in some cases, systemic therapy, in addition to surgical treatment [47,48]. 

The use of adjuvant radiation therapy in patients with cutaneous squamous-cell carcinoma is still a topic of debate, particularly in cases with clear histologic margins, considering the lack of long-term prospective data [49]. A retrospective study gathered information from a total of 508 patients with high-T-stage cSCC and showed that adjuvant radiotherapy after resection with clear margins resulted in a lower 5-year cumulative incidence of both local recurrences (3.6%) and locoregional recurrences (7.5%) than clear margin surgery alone (8.7% and 15.3%, respectively) [50]. However, these data were inconsistent with what other studies have shown [51].

Currently, the guidelines recommend that adjuvant radiotherapy be restricted to patients with compromised surgical margins and extensive peripheral nerve involvement, the involvement of large nerves (≥0.1 mm in diameter) or named nerves, or other high-risk features. There is a dearth of well-conducted, randomized studies evaluating the long-term benefits of this strategy and the magnitude of the survival benefits [8,49].

Patients with cSCC, in general, have a high tumor mutational load due to ultraviolet mutagenesis related to sun exposure, motivating studies using immunotherapy in this histology [52]. In patients with advanced cSCC, the use of cemiplimab, pembrolizumab, and nivolumab has shown objective response rates of around 50%, while also achieving durable disease control and an improvement in the quality of life [23,53,54,55,56]. Some ongoing trials are evaluating the use of pembrolizumab and cemiplimab in an adjuvant setting. However, they face a number of challenges, particularly in respect of patient recruitment (either due to the long period required for recruitment or adequate patient selection) and in defining the duration of treatment [18].

In one study, the use of neoadjuvant cemiplimab, administered at a dose of 350 mg every 3 weeks for up to four doses, was evaluated in 20 patients with resectable stage III or IV SCC, with a complete pathological response in 55% of patients. These results motivated the performance of a phase II multicenter study, which confirmed a benefit in pCR (51%) in those who received neoadjuvant treatment with cemiplimab. In an imaging-based response assessment, defined according to RECIST 1.1 and determined by an investigator’s assessment, the percentage of patients with a complete response was 6%, much lower than that of the pCR, but the reason for this discrepancy is unclear [57].

The results of a phase II study, NEO-CESQ, supported previously presented data, with 38% of patients who received two cycles of neoadjuvant treatment with cemiplimab, followed by surgical resection and then adjuvant treatment for one year, experiencing pCR. There are still immature data not presented in this study [58]. 

As an alternative to cemiplimab, the phase II MATISSE study randomized patients into two groups, one for neoadjuvant treatment with nivolumab and the other for the combination of nivolumab and ipilimumab. Among the 50 patients who received two cycles of the monodrug or combination treatment, a major pathologic response or a clinical complete response was present in 54% and 58%, respectively [59]. Current evidence for the use of neoadjuvant treatment in cCSS is summarized in Table 2.

Currently, no phase three trials determining the best strategy in the neoadjuvant setting are available in the literature.

### 4.3. Basal-Cell Carcinoma

Similar to patients with cSCC, surgery is curative in most cases of BCC, the minority of which are locally advanced or metastatic [60,61]. Most BCCs have genetic alterations in the hedgehog (HH) signaling pathway, leading to the abnormal activation of the pathway and uncontrolled cell proliferation, making it an important therapeutic target [62,63]. 

In 2012, a phase II study, ERIVANCE, evaluated the use of vismodegib in patients with locally advanced and metastatic disease, with the ORR being 60.4% and 48.5%, respectively. After 39 months, the median exposure of patients to vismodegib was 12.9 months, and during this period, discontinuation due to adverse events was the main reason. Muscle spasms, fatigue, and weight decrease were the most common grade 3 and 4 toxicities [64,65]. 

These data encouraged the evaluation of this medication in the neoadjuvant setting. The VismoNeo trial resulted in excellent outcomes, with an ORR of 71% after a mean treatment duration of 6 months, allowing for less extensive and complex surgeries, especially in patients with BCC in functionally challenging locations. Of the 55 patients enrolled, 54 patients (98.2%) experienced at least one adverse event after vismodegib administration, with an average of 6.4 (±3.6) adverse events. Among these patients, 11 had grade ≥3 adverse events (20%). The most frequent adverse events were dysgeusia, muscle spasms, alopecia, fatigue, and weight loss [66].

For patients with locally advanced or metastatic BCC who do not respond to HH pathway inhibitors, the use of Cemiplimab is a second-line option. The approval was based on data from a phase II study that resulted in a 6% complete response and a 25% partial response, indicating clinically meaningful antitumor activity with an acceptable safety profile. A total of 48% of patients had grade 3–4 adverse events, the most common being hypertension and colitis (both occurring in 5% of cases) [21]. 

There is currently little robust evidence in the literature for the use of neoadjuvant treatment in patients with high-risk resectable BCC. However, this approach should be considered on a case-by-case basis, following a careful multidisciplinary discussion.

### 4.4. Merkel Cell Carcinoma

MCC is quite rare and aggressive, and its incidence is increasing worldwide [67]. It is notable for its immunogenicity and its high prevalence in immunodeficiency conditions. MCC exhibits ultraviolet light-mediated mutagenesis and polyomavirus-associated viral carcinogenesis [68]. Immunotherapy plays a central role in the treatment of metastatic MCC, with avelumab being the first immunotherapy approved in this setting [19]. Subsequently, pembrolizumab received accelerated approval in the USA, providing another alternative for these patients [22].

In the localized scenario, wide surgical resection of the tumor and suspicious adjacent lymph nodes is the standard therapeutic approach, with adjuvant radiotherapy reserved for selected cases [69]. However, recurrence rates after surgical treatment are high, around 40% [68]. Based on this, the phase II ADMEC-O study evaluated the use of adjuvant nivolumab 480 mg every 4 weeks for 1 year after surgical treatment. The data were encouraging, showing DFS rates of 85% at 1 year and 84% at 2 years with the use of nivolumab, versus 77% and 73%, respectively, in the observation group, resulting in an absolute risk reduction of 9% (1-year DFS) and 10% (2-year DFS), suggesting a space for immunotherapy in this scenario [19].

Promising results were shown in phase I/II CHECKMATE-358, which evaluated the use of 240 mg of neoadjuvant nivolumab intravenously on days 1 and 15 in patients with resectable MCC, followed by surgery on day 29. Among the 36 patients who underwent surgery, 17 (47.2%) achieved pCR, and approximately one-half of treated patients had tomographic tumor regression, suggesting meaningful antitumor activity with a safety profile of 20% of patients having grade 3 irAEs and 12% discontinuing treatment due to treatment-related AEs. Responses were observed regardless of tumor PD-L1 or TMB status. At a median follow-up of 20.3 months, the median RFS and OS were not reached. The RFS was significantly correlated with the pCR and radiographic response at the time of surgery [70].

Similar to cSCC and BCC, more evidence is required in respect of the effectiveness of neoadjuvant treatments in Merkel cell carcinoma, although the available data are promising. 

## 5. Unanswered Questions

In the context of integrating neoadjuvant immunotherapy into the treatment of individuals with high-risk cutaneous malignancies, a pivotal challenge involves establishing standardized practices and universally accepted criteria for accurately and reproducibly assessing the extent of the pathologic response in resection specimens. Patients undergoing neoadjuvant therapy exhibit varied pathologic responses, influenced by the type of therapy employed (targeted therapy or immunotherapy) [31]. Initial trials defined pCR as the complete absence of a residual viable tumor, patients with a partial pathological response (pPR) as those with up to 50% of the tumor bed occupied by viable tumor cells, and pathologic non-response (pNR) as cases where >50% of the tumor bed is occupied by viable tumor cells [38,45]. Subsequent studies introduced a ‘near pCR/major PR’ category, denoting over 0% but <10% of viable tumor cells. Although newer grading systems propose scoring residual viable tumor cells as a continuous variable, they have not yet attained standardization [70]. The assessment of the pathological response to neoadjuvant therapy lacks homogeneity, relying on the expertise of pathologists predominantly situated in specialized reference centers.

Despite preliminary neoadjuvant trial data indicating a correlation between pCR and improved RFS in patients treated with neoadjuvant immune checkpoint therapy, it remains unclear whether achieving near pCR/major PR also significantly impacts patient outcomes. Hence, it is recommended to document the percentage of residual viable tumors as a continuous variable until more definitive evidence emerges [31].

A crucial matter that needs attention is the need for a more precise selection of eligible patients for neoadjuvant therapy. It is also important to determine the most effective regimen and the duration of the therapy. These clarifications are needed before regulatory agencies can approve neoadjuvant immunotherapy. The absence of such approval is currently limiting the availability of this therapy in clinical practice. Regarding the selection of the type of neoadjuvant therapy employed in melanoma patients (targeted therapy or immunotherapy), no direct comparison is available to back this decision. Despite indirect data suggesting better outcomes for those who received immunotherapy, a randomized trial investigating these two options directly is needed [39].

For patients with up-front surgically resectable diseases, neoadjuvant therapy can lead to disease progression, making surgical treatment impossible. Although rare, studies show that approximately 5% of patients progress to metastatic disease during neoadjuvant treatment [37,71,72]. Another possible disadvantage is adverse events related to the treatment, which may complicate or delay surgical resection, in addition to increasing possible postoperative complications [73]. The incidence of serious events (grade 3 or higher) varied considerably throughout the studies, going from less than 10% to higher than 70%, depending on the population, primary tumor, treatment scheme, and number of cycles administered. Despite the heterogeneity of data, colitis, hypertension, and transaminitis were frequently among the most common irAEs [21,37,38,39,40,70]. 

Being aware of the most common immune-related adverse events (IRAEs) and their chronology is essential for early detection in both preoperative evaluation and postoperative management. Patients who present irAE and are using systemic steroids should undergo surgery after their condition improves and after being weaned off corticosteroids, as, although their use is not a contraindication, it increases the risk of surgical wound complications [73,74]. 

Following neoadjuvant therapy, imaging exams are essential in evaluating the response and therapeutic planning. The radiological response does not always correlate with the pathological response evaluated after surgery. In the OpACIN-neo study, the radiographic response rate per RECIST 1.1 was about 50%, while the pathological response rate was 74% [37,73]. 

Personalizing surgical treatment based on clinical and radiological responses is a major challenge. In good responders, it may be possible to de-intensify surgical treatment, reduce the length of the procedure, and spare them from complete lymph node dissection (CLND), in addition to potentially avoiding the need for adjuvant treatment. However, in poor responders, adopting a strategy that involves lymph node resection and adjuvant treatment seems to be the best option for maximizing outcomes in this subpopulation [75]. 

The PRADO study was designed as an extension of the OpACIN-neo cohort to confirm the response to neoadjuvant immunotherapy, in addition to evaluating the real need for lymph node dissection (TLND) in patients with a major pathological response (MPR) in the largest lymph node. In total, 99 patients were included and treated with at least one cycle of Ipilimumab and nivolumab. The pathological response rate was 72%, 61% had MPR, and an omission of TLND occurred in almost all cases. The 2-year RFS was 93.3% in MPR, with 6% of patients developing regional recurrence and 100% developing distant metastasis-free survival. This showed that treatment de-escalation based on pathological responses, although attractive, is still experimental, suggesting that the further refinement of the patient selection criteria for a limited surgical resection is needed [76]. Although the addition of an adjuvant phase is the rule, there is no clear evidence about its real need. A better understanding of its role remains an area of research to be investigated in future studies. 

Most of these questions mentioned above will be answered over time as more neoadjuvant studies are published and through the analysis of the long-term results of the trials described above.

## 6. Future and Perspectives

The large number of studies in progress and the different strategies being tested mean that, in the near future, new therapeutic options will be available. The identification of biomarkers such as circulating tumor DNA, TMB, and IFN-y can help select the ideal patients for neoadjuvant therapy and be used to guide the best way to personalize treatment, being able to adapt surgery and adjuvant therapy according to each patient’s specific risk and response to therapy. 

Although all the questions regarding neoadjuvant treatment have not yet been fully answered, the data so far support a paradigm shift to considering neoadjuvant treatment as the standard for high-risk skin cancer patients.

## 7. Conclusions

Neoadjuvant therapy presents substantial benefits for patients with locally advanced skin cancer, leading to improved outcomes. Evaluating the pathological response in the resected lesion enables personalized management in respect of surgical and adjuvant treatments. While uncertainties persist about optimal patient selection and dependable biomarkers for treatment responses beyond pathological assessment, the encouraging results suggest that neoadjuvant therapy might soon be regarded as a standard approach for selected high-risk skin tumors—notably, melanoma.

## Figures and Tables

**Table 1 medsci-12-00035-t001:** Pathological response and safety of neoadjuvant ICI arms in clinical trials of melanoma.

Trial	Treatment	pCR (%)	MPR (%)	pPR (%)	pNR (%)	AE G1/2 (%)	AE G3/4 (%)
OpaCIN-Neo	Ipi 3 + Nivo 1 ×2	47	70	10	20	57	40
	Ipi 1 + Nivo 3 ×2	57	64	13	23	77	20
	Ipi 3 + Nivo 3 ×2	23	46	19	38	50	50
NCT02519322	Nivo 3 ×3	25	-	-	-	92	8/0
	Ipi 3 + Nivo 1 ×3	45	-	-	-	91	73/0
NCT02519322	Nivo + Rela ×2	57	64	7	27	94	26
INMC	Nivo ×4 Pembro ×1	20	25.7	8.6	65.7	NR	NR
	Combo ICI	42.7	61.2	13.6	25.2	NR	NR
NeoTrio	Pembro ×2	30	40	15	35	-	30
	Pembro ×2 -> Dabrafenib + Trametinib 1 week	20	30	20	50	-	25
	Pembro ×2 + Dabrafenib + Trametinib 1 week	50	55	25	20	-	55
NCT02434354	Pembro ×1	19	11	-	70	-	<30
PRADO	Ipi 1 + Nivo 3 ×2	49	61	11	21	75	22
SWOG S1801	Pembro ×3	38	53	21	21	NR	7
NeoPele	Pembro ×2 + Lenvatinib	40	55	20	25	NR	45
NCT04303169	Pembro + Vibostolimab	38	50	31	19	92	8
	Pembro + Gebasaxturev	28	40	12	48	84	24
	Pembro ×1	40	47	27	26	80	7
NCT02231775	Dabrafenib/Trametinib	58	-	17	25	92	15
NCT01972347	Dabrafenib/Trametinib	49	-	51	0	100	26/3
NADINA	Ipi 80 mg + Nivo 240 mg ×2	11.8	59	8	28.8	NR	29.7

Abbreviations: ICI, immune checkpoints inhibitors; Ipi 1, Ipilimumab 1 mg/kg; Ipi 3, Ipilimumab 3 mg/kg; Nivo 1, Nivolumab 1 mg/kg; Nivo 3, Nivolumab 3 mg/kg; Rela, Relatlimab; Pembro, Pembrolizumab; pCR, complete pathological response; MPR, major pathological response; pPR, partial pathological response; pNR, pathological no response; AE, adverse effects.

**Table 2 medsci-12-00035-t002:** Summary of the pathological response of neoadjuvant ICI arms in clinical trials.

Trial	Treatment	pCR (%)	MPR (%)	pPR (%)	pNR (%)	AE G1/2 (%)	AE G3/4 (%)
NCT04154943	Cemiplimab 350 mg q3w ×4	51	64	11	25	87	18
MATISSE	Ipi 1 ×1 + Nivo 3 ×2	NR	45	10	45	NR	4
	Nivo 3 ×2	NR	50	30	20	NR	8
NEO-CESQ	Cemiplimab 350 mg q3w ×2	39	47	4	48	57	30

Abbreviations: ICI, immune checkpoints inhibitors; Ipi 1, Ipilimumab 1 mg/kg; Nivo 3, Nivolumab 3 mg/kg; q3w, every 3 weeks; pCR, complete pathological response; MPR, major pathological response; pPR, partial pathological response; pNR, pathological no response; AE, adverse effects.

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
