# Peer review of "Emerging Indications for Neoadjuvant Systemic Therapies in Cutaneous Malignancies"

_medsci, 2024, doi:10.3390/medsci12030035_

Round 1

Reviewer 1 Report (Previous Reviewer 3)

Comments and Suggestions for Authors

The manuscript is now acceptable for publication.

Author Response

Thank you very much for taking the time to review this manuscript.

There are no reviewer suggestions for adjustments.

We would again like to thank the reviewers for reviewing our manuscript and taking the time to do so.

Reviewer 2 Report (Previous Reviewer 2)

Comments and Suggestions for Authors

The authors have revised the manuscript to address my concerns.  Nothing to add.  Manuscript does not require further revisions

Author Response

Thank you very much for taking the time to review this manuscript.

There are no reviewer suggestions for adjustments.

We would again like to thank the reviewers for reviewing our manuscript and taking the time to do so.

Reviewer 3 Report (New Reviewer)

Comments and Suggestions for Authors

The work has scientific relevance, but minor adjustments must be made. 

1-When we talk about literature review, we describe the methodology used. 

 I believe the study methodology should be added: date of choice of articles, last 10 years, 5 years? What were the descriptors used, their combination, and the database used?  How many articles were found, and what were the inclusion and exclusion criteria?  

2-In the work, some parts are referenced as adverse effects, but this point is not explored at any point in the work. Describe the main adverse effects of each neoadjuvant. 

 3- Table 1 should be adjusted

4- Minor typos should be adjusted

Author Response

Thank you very much for taking the time to review this manuscript. Please find the detailed responses below and the corresponding revisions/corrections highlighted (in yellow) in the re-uploaded files.

Point-by-point response to authors' comments and suggestions

Comments 1: When we talk about literature review, we describe the methodology used.

Answer 1: Thanks for pointing this out. We agree with this comment. We have added the methodology used, item 2 of the manuscript, between lines 106 and 117, pages 3 and 4. It is worth noting that our manuscript is an opinion article and not a systematic review, but we believe it is extremely important to detail the methodology used for our search in databases.

Comments 2: In the work, some parts are referenced as adverse effects, but this point is not explored at any point in the work. Describe the main adverse effects of each neoadjuvant.

Response 2: We agree with the comment. We made changes to the manuscript, detailing the main adverse events related to neojuvant treatment. They are in the following excerpts: lines 185 to 187 and 189 to 190 of page 5; 342 to 343 and 356 to 357 on page 10; 382 to 384 on page 11 and 427 to 431 on page 11.

Comments 3: Table 1 should be adjusted

Response 3: We agree with the comment and the adjustment in table 1 was made as advised in the manuscript.

Comments 4: Minor typos should be adjusted

Response 4: We agree with the comment and made the adjustments highlighted in the manuscript as per the guidance.

We would again like to thank the reviewers for reviewing our manuscript.

This manuscript is a resubmission of an earlier submission. The following is a list of the peer review reports and author responses from that submission.

Round 1

Reviewer 1 Report

Comments and Suggestions for Authors

Thank you for the opportunity to review this review on neoadjuvant systemic therapies for skin cancers.

The manuscript could benefit if further focused on neoadjuvant therapies, that can be more clearly presented, as it currently presents a discussion of adjuvant and neoadjuvant therapies together.

Some further suggested clarifications,

-in the Introduction, could you add citations for evidence that adjuvant treatment for high-risk cSCCs with clear surgical margins minimizes recurrence?

-in section 2. Comparing neoadjuvant theray to adjuvant therapy and saying that it holds advantages- could you present some data comparing these approaches in patients?

-the references 43,44, cited for the use of adjuvant systemics for high-risk cSCC are not correct  (page 7)

-could you clarify if there is a randomized study mentioned, showing the long-term benefits of adjuvant RT for high-risk cSCC?

- is there a guideline recommendation that adjuvant RT should be used for high-risk cSCC?

-could you detail the objective response rates of immunotherapy for cSCC?

--the references seem not to be cited correctly. Page 8, for NEO-CESQ study , the cited reference 50 is wrong-Gross et al. 2022.

--page 8, “..use of neoadjuvant treatment in patients with high-risk resectable BCC, yet this approach can be considered in a case-by-case approach, following a careful multidisciplinary discussion.” Could you cite some references supporting this suggestions?

-please revise to improve the English language if possible.

Comments on the Quality of English Language

could be improved if possible

Reviewer 2 Report

Comments and Suggestions for Authors

Lins Junior et al. provide a well written and timely review of the emerging data for the role of neo-adjuvant treatment of cutaneous malignancy.  They provide a detailed review of published trials and excellent discussion.  I think this paper is worthy of publication with some minor revisions.  I would ask that the authors address:

1)  The authors provide excellent discussion of the SWOG s1801 data.  However, it is my understanding that there was a significant difference in the reported rates of pCR in the central pathology review versus the local pathology review.  I think this is an important issue to raise to highlight the need for standardization of pathology practices and the challenges of using pathological response as a prognostic factor - the authors do this in the "Unanswered questions section" but suggest add the example of the SWOG study discrepancy

2) The authors mention that in the cSCC section there was a discrepancy between the pathological and radiological response.  This discrepancy was also seen on the Prado trial and the OpaCin Neo trial.  The authors should add some discussion of this to the text - they do discuss in the "Unanswered questions section" but perhaps could add some more discussion as to the potential reasons for this discrepancy

Minor:

1) "signals" is spelled wrong on page 3 line 107

Reviewer 3 Report

Comments and Suggestions for Authors

A review by Domingos Lins Junior et al summarizes the rationale, evolving strategies, currently available evidence, and challenges regarding neoadjuvant therapy in cutaneous malignancies.

Specific comments:

1. "Emerging indications" in the title should be replaced with more clear message.

2. The manuscript should be revised with regard to consistency. For example, the Authors write IFN-gamma (line 132) once, while in another sentence they write IFN-y (line 197) etc.

3. Overall, the manuscript summarize selected clinical trials with little or no substantially novel general directions to be indicated. In my opinion, the Authors should put this review in more wide context and possible future perspectives.

Comments on the Quality of English Language

minor revision required